# A Novel Underwater Acoustic Target Recognition Method Based on MFCC and RACNN

**DOI:** 10.3390/s24010273

**Published:** 2024-01-02

**Authors:** Dali Liu, Hongyuan Yang, Weimin Hou, Baozhu Wang

**Affiliations:** 1School of Electronics and Information Engineering, Tiangong University, Tianjin 300387, China; 2130070820@tiangong.edu.cn; 2School of Information Science and Engineering, Hebei University of Science and Technology, Shijiazhuang 050018, China; wangbz@hebust.edu.cn

**Keywords:** UATR, ship-radiated noise, MFCC, residuals mechanisms, attention mechanisms

## Abstract

In ocean remote sensing missions, recognizing an underwater acoustic target is a crucial technology for conducting marine biological surveys, ocean explorations, and other scientific activities that take place in water. The complex acoustic propagation characteristics present significant challenges for the recognition of underwater acoustic targets (UATR). Methods such as extracting the DEMON spectrum of a signal and inputting it into an artificial neural network for recognition, and fusing the multidimensional features of a signal for recognition, have been proposed. However, there is still room for improvement in terms of noise immunity, improved computational performance, and reduced reliance on specialized knowledge. In this article, we propose the Residual Attentional Convolutional Neural Network (RACNN), a convolutional neural network that quickly and accurately recognize the type of ship-radiated noise. This network is capable of extracting internal features of Mel Frequency Cepstral Coefficients (MFCC) of the underwater ship-radiated noise. Experimental results demonstrate that the proposed model achieves an overall accuracy of 99.34% on the ShipsEar dataset, surpassing conventional recognition methods and other deep learning models.

## 1. Introduction

Underwater acoustic target recognition (UATR) has always been one of the main areas of research in underwater technology. It is commonly applied for marine biological surveys, ocean exploration, and underwater scientific activities [1,2]. However, the mechanism of radiated noise generated by underwater targets is very complex, and the radiated noise contains multiple components, including continuous spectrum components and strong discrete spectrums. Furthermore, many factor such as spatiotemporal variations in the underwater acoustic channel, multipath effects, and Doppler effects can all affect the propagation of underwater acoustic signals [3,4,5]. As a result, UATR is a highly challenging and difficult technology.

The recognition of underwater target-radiated noise can be divided into two steps: feature extraction and recognition algorithms. Scholars have been trying for decades to artificially extract features from underwater acoustic targets [6,7,8], and the methods include short-time Fourier transform (STFT) [9], low frequency analysis and recording (LOFAR) [10], Mel-frequency spectrum [6], demon noise envelope modulation detection (DEMON) [11], and Mel Frequency Cepstral Coefficients (MFCC) [12]. Traditional algorithm, such as GMM [13] and SVM [14], were used for the underwater acoustic field. These manually extracted features and algorithms have played a significant role in the underwater acoustic field, for example, MFCC features were extracted in a lot of work [15,16,17] for UATR. Xin et al. note in the classification of ship-radiated noise signals that the traditional classifier has many limitations. In the ocean, the complex environmental noise, the low SNR of underwater acoustic signals, and the interference of other ship-radiated noise make it difficult for the traditional classifier to obtain a good recognition effect [18]. Q et al. proposed the design of a real-time and accurate underwater target classifier method using localized wavelet acoustic pattern (LWAP) and multilayer perceptron (MLP) neural network, which requires expertise and experience [19]. Therefore, it is necessary and full of significance to invent an algorithm that can automatically learn from the data or the extracted features and categorize underwater targets without specialized knowledge.

The rapid development of deep learning has had a significant impact on research from various fields [20,21]. Neural units in deep neural networks can imitate the nonlinear decision-making process of human neurons, which allows the network to fit samples of any distribution while maintaining good robustness and accuracy. In the field of deep learning, residual and attention mechanisms are two important concepts. The residual mechanism is used to solve the problem of gradient vanishing in deep neural network training. In addition, the attention mechanism is raised to improve the model’s attention to the input sequence. The residual structure is designed to facilitate the network in learning a constant mapping.

This means that the input is passed directly to the output, simplifying the learning process. The direct input-to-output passage aids in learning the remainder of the information. This residual learning approach proves beneficial for handling complex tasks. Moreover, it enhances the network’s capability to learn abstract features effectively. By dynamically assigning different weights to different parts, the attention mechanism allows the model to handle input sequences of different lengths and improves the generalization ability of the model. Neural networks are capable of building sufficiently complex architectures in terms of depth and width. Compared to traditional manually designed feature extraction methods, adaptive feature extraction based on deep neural networks demonstrates better performance due to its better generalization properties and lower requirements for factors such as signal-to-noise ratio and sample distribution. Therefore, the deep learning network algorithm is highly accurate and robust in UATR. Liu et al. [21] proposed three-dimensional fusion features and performed UATR using a ResNet18 network. Wang et al. [11] designed a one-dimensional convolutional neural network (1D-CNN) and assessed its accuracy by designing datasets containing various Doppler shifts and signal-to-noise ratios. The experimental results showed that the 1D-CNN model outperformed the traditional Machine Learning (ML) model. In 2013, IBM researcher Brian Kingsbury et al. [22] used log Mel filter coefficients as inputs to a deep convolutional network to further extract “raw” features (Mel filter coefficients), and experiments have shown that the recognition rate is relatively improved by 13–30% compared to the traditional Gaussian mixture model and by 4–12% compared to the deep neural network. Wang et al. [23] proposed a UATR method that combines multi-dimensional fusion features and an improved deep neural network (MFF-MDNN). The network utilized the GMM to correct the structure of the deep neural network. When the datasets contained weak underwater background noise, the proposed method achieved a recognition accuracy of 94.3%. However, the MFF-MDNN method incurred high computational costs due to the extraction of multi-dimensional fusion features. Van and T [24] proposed a method for UATR using a dense CNN model. This approach yielded an overall accuracy of 98.85% at 0 dB SNR.

Traditional methods for underwater target identification often grapple with the inherent influence of human factors, leading to fluctuating levels of accuracy and timeliness in identification outcomes. Various recent modern feature extraction and recognition algorithms need to be improved in terms of accuracy, recognition speed, and network volume. Addressing these challenges, in this article we propose a Residual Attentional Convolutional Neural Network (RACNN), which employs CNN modules, attention mechanisms, and residual techniques. The main contributions of this paper are as follows:(1)With the application of residual and attention mechanisms, we enhance the learning capability, fault tolerance, and emphasis on vital information of networks. This facilitates the suppression of various environmental noises, the extraction of deep abstract features of the signal, and the improvement of sensitivity to critical information.(2)Compared to other networks, we reduce the number of parameters and effectively highlight crucial information hidden in the time–frequency spectrum. This leads to a reduction in the use of computational resources and an increase in computational efficiency and speed, which makes sense in practical applications.

The structure of this article is as follows; Section 1 is an introduction to the research background of underwater acoustic technology and related technologies. Section 2 describes the characterization and extraction of raw acoustic signals and the design of deep neural networks. Section 3 gives specific experimental results and analysis. The fourth part summarizes the contents of the whole paper and the outlook of future technological progress.

## 2. Proposed Method

This section mainly consists of two parts. Part A introduces the MFCC feature extraction process. In this segment, we delve into the details of how the MFCC features are methodically extracted, shedding light on the underlying processes and methodologies. Part B introduces the network structure of the proposed RACNN. This subsection provides a detailed account of its structural components, layers, and mechanisms.

### 2.1. MFCC Feature Extraction

The problems faced in directly recognizing radiated noise from underwater targets are noise interference, spectral time variability, and frequency attenuation. In order to overcome these problems, the Mel Frequency Cepstrum Coefficient (MFCC) is used as the feature representation. The MFCC has the following advantages: (1) Removal of noise interference: the MFCC filters and normalizes the spectrum in the feature extraction process, which can attenuate the interference of noise on the target-radiated noise to a certain extent. (2) Resistance to temporal variability: the MFCC compresses the spectrum using Discrete Cosine Transform (DCT), which captures the main features of the spectrum and reduces the frequency attenuation. Transform (DCT) to compress the spectrum, which can capture the main features of the spectrum and reduce the effect of time variability. (3) Reduced feature dimension: MFCC converts the spectrum into cepstrum coefficients, which effectively reduces the dimension of the features and makes the subsequent pattern recognition and classification tasks more efficient.

MFCC is a widely used feature in the field of speech and voice recognition, which was first introduced by Davis and Mermelstein in 1980 [25]. MFCC feature is an efficient tool for describing and analyzing ship-radiated noise. It achieves this by simulating the characteristics of human ear perception, compressing spectral information, reducing dimensionality, and facilitating accurate recognition [26]. The conversion formula between linear frequency and Mel frequency is as follows:(1)$$\mathrm{Mel}(f)=2595\times \mathrm{lg}(1+\frac{f}{700})$$
where Mel(*f*) denotes the Mel frequency and *f* denotes the Fourier frequency.

The extraction of MFCC entails a series of critical steps. Initially, the technique initiates with framing and windowing applied to a time-domain signal, segmenting it into discrete frames. Following this, Fourier transformation is meticulously applied to each frame signal, resulting in the generation of the local spectrum. A pivotal stage in the MFCC extraction process involves the mapping of data from the Fourier domain onto the Mel scale. This transformation is achieved through the utilization of Mel-scale filter banks, imparting a characteristic spectral shape that mirrors the non-linear human auditory system. The transformation equation is as follows:(2)$$H_{m}(k)=\left\{\begin{array}{ll} 0\; & ,\mathrm{if}\;k<f(m-1)\\ \frac{2(k-f(m-1))}{\left(f(m+1)-f(m-1))(f(m)-f(m-1)\right)} & ,\mathrm{if}\;f(m-1)<k<f(m)\\ \frac{2(f(m+1)-k)}{\left(f(m+1)-f(m-1))(f(m)-f(m-1)\right)} & ,\mathrm{if}\;f(m)<k<f(m+1)\\ 0\; & ,\mathrm{if}\;k>f(m+1)\; \end{array}\right.$$
where $H_{m}(k)$ represents the frequency response of the *m*-th Mel-scale filter and *k* denotes the frequency index of Discrete Fourier Transform (DFT). f(m) represents the value of the Fourier frequency corresponding to the *m*-th Mel filter.

Each filter is multiplied and summed with the Fourier transform of the signal, resulting in a set of logarithmic energy values, which is as follows:(3)$$s(m)=\ln(\sum_{k=0}^{N-1}\left|X_{a}(k){|}^{2}H_{m}(k\right)),0\leq m\leq M$$
where s(m) represents the *m*-th logarithmic energy value, $|X_{a}(k){|}^{2}$ represents the power spectrum of the signal, which can be obtained by DFT, and $H_{m}(k)$ represents the frequency response of the *m*-th Mel-scale filter.

Finally, to finalize the extraction of MFCC, Discrete Cosine Transform (DCT) is applied to the s(m). The above extraction process is shown in Figure 1.

The paper performs MFCC feature extraction on experimental samples of duration 1 s. The frame length FFT of the data subframes is 2048 points, and the neighboring frames overlap 1536 points. The number of Mel filters is set to 50, and a total of 44 groups of MFCC data are obtained, which are spliced into a two-dimensional array in chronological order to obtain the MFCC data as shown in Figure 2 below, as a sample for underwater target recognition. This sample is flattened before inputting into the network, and the data length is 2000.

### 2.2. Design of Deep Learning Networks

The input data of the proposed RACNN model consist of the MFCC feature of the radiated noise from underwater targets. These coefficients are flattened along the dimension of the cepstral coefficients to form a one-dimensional sequence (i.e., 1D-MFCC). The conceptualization of the RACNN model was informed by the architectural principles of both ResNet and Attention Model networks. The schematic representation of the proposed RACNN architecture is elucidated in Figure 3, delineating its composition into distinct segments. The model comprises four principal components: three Block_A modules, two Block_B modules, a Fully Connected layer, and a Softmax layer. Each element contributes uniquely to the network’s ability to process and understand complex patterns within the input data.

The core functionality of the RACNN unfolds as follows: I network takes a 1D-MFCC sequence as input, a representation of the acoustic features derived from underwater signals. Through the intricate interplay of the defined modules and layers, the network produces an output that corresponds to the category associated with the maximum predicted probability.

In deep learning networks, the convolutional layers extract features from the input data. Each convolutional layer consists of multiple convolutional kernels, where each kernel corresponds to a set of weight coefficients and a scalar bias. Given an input sequence x and the weight vector of the convolutional kernel w, the *i*-th element of the output y is defined as follows:(4)$$y[i]=\sum_{j=0}^{k-1}\left(w[i]\cdot x[j]\right)+b_{i}$$
where w[j] represents the *j*-th element of the weight vector *w*, *x*[*j*] represents the *j*-th element of the input sequence *x,* and $b_{i}$ is the bias.

In the Block_A in Figure 1, the first layer is Conv_1, which is a convolutional layer with a 1 × 3 kernel size, 1 × 1 stride size, and the activate function Elu, which is as follows:(5)$$\mathrm{Elu}(x)=\left\{\begin{array}{ll} x, & x\geq 0\\ e^{x}-1, & x\;<\;0 \end{array}\right.$$

Conv_1 serves two primary purposes: to map low-dimensional input sequence data to high-dimensional representations and to extract abstract features of the input signal. The number of channels in the Conv_1 layer of the first Block_A is set to 256, and the number of channels in the Conv_1 layer of each subsequent Block_A is half that of the previous Block_A. The output of Conv_1 is passed into the Batch Normal (BN) layer, which normalizes and standardizes the data. This process helps accelerate the convergence speed of the network. The following layer is the Max Pooling (MP) layer, which is responsible for downsampling the data. However, the MP layer is replaced by the Average Pooling (AP) layer in the latter two Block_A. After the MP layer, two channels are applied. In the upper channel, a convolution layer Conv_2 with a 1 × 1 kernel size is employed, while a convolution layer Conv_3 with a 1 × 3 kernel size is used in the lower channel. Convolutional kernels of different sizes are able to capture representative features from different scales. The output results of the two channels are concatenated along the channel axis, and the obtained data are added to the MP output data to form a residual structure.

The input vector of Block_A is represented by $p\in R^{1\times 1\times W}$, the output of Conv_1 is represented by $p^{\prime }\in R^{C\times 1\times W}$, and the output vector by MP layer is P″. The output vector P′ and P′′ and the output result of Block_A can be described as follows:(6)$$\begin{array}{l} P\prime =\mathrm{eLu}(f_{n}^{1\times 3}(P))\\ P^{\prime \prime }={\mathrm{MaxPool}}_{4}[\mathrm{Batchnormal}(P\prime )]\\ A_{out}=\mathrm{Add}(\mathrm{Concate}[Conv\_2(P^{\prime \prime }),Conv\_3(P^{\prime \prime })],P^{\prime \prime }) \end{array}$$
where Elu represents the activation function, $f_{n}^{1\times 3}$ represents that the number of convolution kernels is 3, and the size of the convolution kernel is 1 × 3, and ${Maxpool}_{4}$ represents a pooling layer with a size of 4.

The initial layer of Block_B in Figure 3 consists of a convolution layer with 1 × 3 kernel size, followed by a BN layer. The output of this layer is fed into the SE_Block, which employs automatic learning to determine the significance of feature maps in various dimensions. This results in a weight matrix that is multiplied by the feature maps. Channels with a higher weight value indicate greater importance, leading to higher correlation with classification recognition. SE_Block demonstrates the implementation process of the attention mechanism. The implementation process of the SE mechanism is shown in Figure 3, where the numbers of N and C are set to 1/16 of the number of input feature map channels and the number of input feature map channels, respectively. When the data pass through the SE module, each channel of the input feature map is multiplied by a weight factor, which enhances the channels with representative feature maps and reduces the importance of the data of the unimportant channels, which in turn reduces the network’s attention to them. The output of the SE_Block is then fed through a convolutional layer and added to the output of the first convolutional layer of Block_B, serving to prevent overfitting and further fit the signal feature.

The subsequent module of Block_B is an FC layer. The input of the FC layer is the flattened output of the convolution layer, the output data include a 1 × 256 vector, and the activate function is Relu. It realizes the end-to-end learning process by unfolding the multi-channel data output from the convolution and mapping it into one-dimensional vectors through a matrix. The concluding layer in the network architecture is the Softmax layer, designed to map the collective outputs of multiple neurons into a probabilistic range between 0 and 1. This critical layer serves to normalize the output data generated by the neurons, ultimately selecting the output with the highest probability as the final prediction. The Softmax function, pivotal in achieving this normalization and probability assignment, is expressed as follows:(7)$$S(z_{i})=\frac{e^{z_{i}}}{\sum_{n}e^{z_{j}}}$$
where $S(z_{i})$ represents the probability generated by the *i*-th neuron after passing through the Softmax layer, *n* denotes the total number of output neurons, and $z_{i}$ represents the value of the *i*-th neuron before normalization through the Softmax layer.

## 3. Experiments and Results

### 3.1. Experiment Setup and Dataset

In the course of this study, the dataset for training and validation is derived from the ShipsEar [27] dataset. This comprehensive dataset encompasses sound recordings originating from various vessels, capturing a spectrum of ship-radiated noise levels along the Atlantic coast of Spain during the years 2012 and 2013. Within the dataset, a meticulously curated collection of 91 samples is presented, comprising a diverse array of acoustic scenarios. This dataset encapsulates 11 distinctive types of ship-radiated noise, each offering unique insights into the acoustic landscape. The categories of propagated noise include natural environmental sounds, fish boats, trawlers, mussel boats, tugboats, dredgers, motorboats, pilot boats, sailboats, passenger ships, ocean liners, and ships. Additionally, the dataset incorporates a category dedicated to background noise, providing a comprehensive representation of the acoustic environment prevalent along the Atlantic coast during the specified time frame. The inclusion of these varied noise sources ensures the dataset’s richness and relevance to the study’s objectives, facilitating a robust evaluation of the proposed RACNN model in the context of underwater target recognition.

The 11 types of ship-radiated noise were classified into 4 categories. Background noise was designated as a separate category. In total, there are five categories in the classification results, as follows:

Class A: Natural Ambient Noise.

Class B: Fishboat, Trawler, Mussel Boat, Tugboat, Dredger.

Class C: Motorboat, Pilot Ship, Sailboat.

Class D: Passengers.

Class E: Ocean Liner, Roro.

The FFT spectrum, the result of Mel filter and the DTC transformation of five types of noise are shown in Figure 4.

The 91 audio recordings from the ShipsEar dataset were split into 1 s intervals to generate MFCC features of the underwater targets as experimental samples. The samples were then randomly split into a training set and a testing set in a 0.8:0.2 ratio. Specifically, the training set consisted of 9040 samples, while the testing set comprised 2260 samples. As show in Table 1, of these 9040 training samples, 912 belonged to category A, 1500 to category B, 1248 to category C, 3416 to category D, and 1964 to category E. The 2260 samples in the test set contain 288 belonging to class A, 375 to class B, 312 to class C, 854 to class D, and 491 to class E. The test set is a collection of 2460 samples.

During training, the RACNN model utilized the Adam optimizer with a learning rate of 0.001, a training batch size of 128, and a training period of 60 epochs. The learning rate decay strategy was applied to RACNN, decreasing the learning rate to 0.2 every 20 epochs.

### 3.2. Experiment Results and Discussion

In the RACNN model, Block_A incorporates multi-channel and residual connection mechanisms to enhance feature extraction and improve the convergence rate of the data, as well as to avoid gradient vanishing and explosion issues. Block_B features residual connection and attention mechanisms to prioritize critical information for classification and recognition. The number of Block_A and Block_B, as well as the units of the FC layer, can impact the performance of the RACNN. To evaluate the impact of each block and determine the optimal parameters for the RACNN model, we proposed a series of networks with different configurations. The parameters and performance of the six typical networks, namely Model_1 to Model_6, are presented in Table 2 and Figure 5. The experimental results show that by testing on the ShipsEar dataset, the best accuracy of the model validation set can be achieved as 0.9934, while the network parameters are 149 K and the computational FLOPs(G) is 0.139.

When the number of FC units is equal to 0, Model_1 has the lowest recognition accuracy, while Model_5 has the highest accuracy. This indicates that the presence of more Block_A and Block_B allows for more representative features to be extracted and enhances the ability of the network to recognize underwater targets. Notably, Model_5 exhibits a higher accuracy rate of 1.6% compared to Model_3. However, it also has an additional 20 K parameters. Moreover, the presence of an FC layer can significantly increase recognition accuracy. Specifically, for Model_4, an FC layer with 256 units yielded a 2.9% improvement in accuracy compared to Model_3. It should be noted that enhancing performance by increasing the units in Block_A, Block_B, and FC is not always feasible. Once the scale of network parameters exceeds a certain threshold, the performance may decrease. This may be because excluding the first Block_A, we will reduce the number of channels of the feature map to half when passing through other blocks, which will result in some important information missing after multiple operations. As an illustration, the Model_6 model had a recognition accuracy decrease of 1.2% when an FC layer of 64 units was added, compared to Model_5 without an FC layer. In the following experiments, we choose Model_4, which has the best performance as the RACN model.

In deep learning, the size of the convolutional kernel and the choice of activation and cost functions have a great impact on the overall performance of the network. Activation functions are very important for artificial neural network models to learn deep features of the signal and to understand very complex nonlinear mapping relationships. The size of the convolution kernel affects the ability of the network to capture information and the number of parameters. In Table 3, we compare several classical activation functions, i.e., Relu, Sigmoid, and Elu functions. In addition, we compare the recognition accuracy of the network when using different sizes of convolutional kernels. The data in the table show that at constant convolution kernel size, i.e., when the size of the parameters of the network is constant, the network achieves the best recognition result of 0.9934 using the Elu activation function, which is 1.42% better than the Sigmoid function, and 0.31% better than the Relu function. The Sigmoid function is saturated when the variables take on very large positive or negative absolute values, which means the function becomes flat and insensitive to small changes in the input.

The Relu activation function, compared to the Elu function, in the negative region of the derivative transfer will be killed; this phenomenon is called dead Relu, and this will lead to the corresponding parameters never being updated. The size of the convolutional kernel directly affects the number of parameters of the network, which further affects the computation rate. Experimental comparisons show that when the convolutional kernel size is 3, the network not only has the smallest number of parameters but also has the highest recognition accuracy. When the convolutional kernel size is 5 or 7, the network might be unable to capture a small range of features, resulting in the loss of important information and a decrease in accuracy.

The training process of the best RACNN model is shown in Figure 6, which showcases the rapid convergence of the model.

The recognition result of the underwater targets test dataset is shown in Figure 7. Each column of the confusion matrix represents the predicted label category, and each row represents the true category of the sample.

In addition, we use accuracy, *Recall*, and F1-score as performance metrics to characterize the performance of the methodology, and each performance metric is calculated as follows.
(8)$$\begin{array}{c} Precision=\frac{TP}{\left(TP+FP\right)}\\ Recall=\frac{TP}{\left(TP+FN\right)}\\ F1\text{-}score=2\cdot \frac{Precision\cdot Recall}{Precison+Recall} \end{array}$$
where TP denotes True Positive, FP denotes False Positive, and FN denotes False Negative.

The performance metrics, such as accuracy, recall, and *F*1-*score*, can be obtained from the confusion matrix, which are presented in Table 4. The experiment resulted in a lowest recall rate of 0.9918, a lowest accuracy of 99.36%, and a lowest *F*1-*score* of 0.9418. The average recall, accuracy, and *F*1-*score* were 0.9944, 0.9945 and 0.9945, respectively. The experimental results reveal that the proposed RACN model can recognize underwater acoustic targets with good accuracy.

In order to further evaluate the performance of the proposed RACNN model, we applied some classical deep learning classification networks to recognize underwater targets, such as Vgg16 [28] and Resnet34 [29], and compared the performance of these networks. After these networks were modified to accept underwater target data as input, they were trained using randomly initialized parameters. The experiment results are shown in Table 5, and the training process is shown in Figure 8. The experiment process indicated that RACNN began to converge after approximately 10 iterations due to the incorporation of residual and attention structures. Other networks required around 20 iterations to stabilize. Compared with Vgg16 and ResNet34, the RACN network achieved the highest accuracy of 0.9934, which is 1.04% and 0.22% better than Vgg16 and Resnet34 networks, respectively.

Additionally, all the networks achieve a recognition accuracy of over 0.98 for the dataset, indicating the effectiveness of the MFCC feature extraction method. The RACNN has the fewest parameters compared with Vgg16 and ResNet34 networks, whose parameters are 33 M and 21 M, respectively. This means that the RACNN consumes less resources and computing power, achieving faster calculation speed.

We performed a comprehensive comparative analysis by juxtaposing our RACNN network with various other Underwater Acoustic Target Recognition (UATR) methodologies proposed by researchers in the field. The summarized findings are presented in Table 6, offering a clear overview of the performance metrics across different classification methods. Upon meticulous evaluation, it is evident that the fusion of MFCC and RACNN emerges as the standout performer, achieving the highest accuracy while maintaining a notably compact parameter size. Notably, our network exhibits superior recognition performance, showcasing not only the highest accuracy but also the fewest Floating Point Operations (FLOPs) and parameters.

These results underscore the effectiveness of the RACNN network in learning distinctive features from underwater acoustic signals in a remarkably efficient manner. The combination of advanced feature extraction methods and the inherent structure of the network contributes to the network’s ability to discern and exploit unique characteristics within the acoustic data. The outcome suggests that RACNN excels in capturing and leveraging relevant information, ultimately leading to enhanced recognition performance within a concise parameter space. The results not only affirm its superior performance compared to existing methodologies but also underscore its potential for real-world applications where efficiency and accuracy are paramount considerations.

## 4. Conclusions

Traditional methods for identifying underwater target-radiated noise have typically relied on limited features, making them less effective in complex sea conditions. This paper introduced the RACNN model, utilizing MFCC data as input for the recognition of underwater ship-radiated noise. The proposed network was trained and assessed for performance using the ShipsEar dataset. Integrating residual and attention mechanisms, the network combines the strengths of ResNet and Attention models. By leveraging the CNN convolutional module, the RACNN model facilitates self-learning of target features, mitigating issues such as gradient dispersion and vanishing. Simultaneously, the network effectively extracts valuable information, discarding redundant data and prioritizing channels with representative features, thereby enhancing accuracy. Experimental results demonstrate that RACNN achieves an impressive recognition accuracy rate of 99.34%. This surpasses the accuracy and time efficiency of models such as VGG and ResNet. Furthermore, RACNN proves adept at extracting high-level information conducive to classification, showcasing significant potential for underwater target recognition.

This model outperforms other UATR models in both accuracy and parameter scale. In light of these findings, future work will explore avenues for refining and extending the RACNN model, considering additional datasets and further optimizing its performance in diverse underwater scenarios. This ongoing research aims to contribute to the continuous advancement of underwater acoustic target recognition methodologies.

The RACNN model presents a robust and efficient solution for underwater target recognition, demonstrating superior performance and holding promise for applications in various maritime environments.

## Figures and Tables

**Figure 1 sensors-24-00273-f001:**
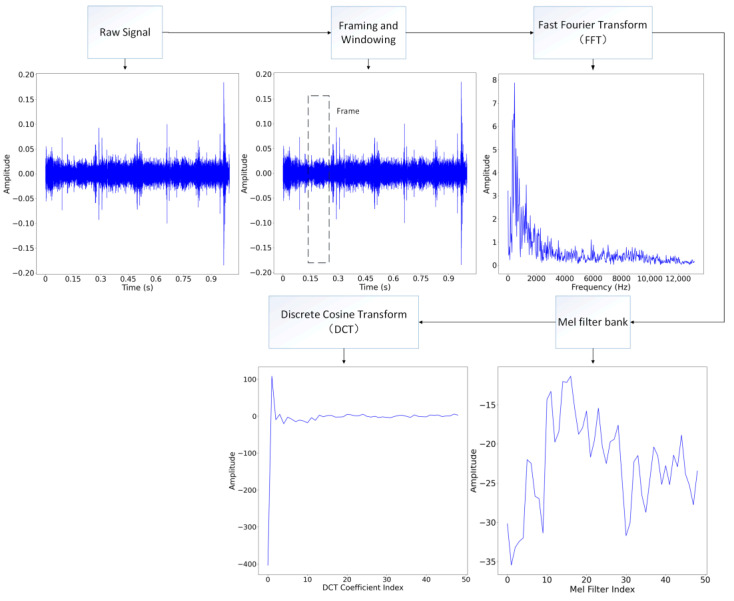
The extraction process of MFCC.

**Figure 2 sensors-24-00273-f002:**
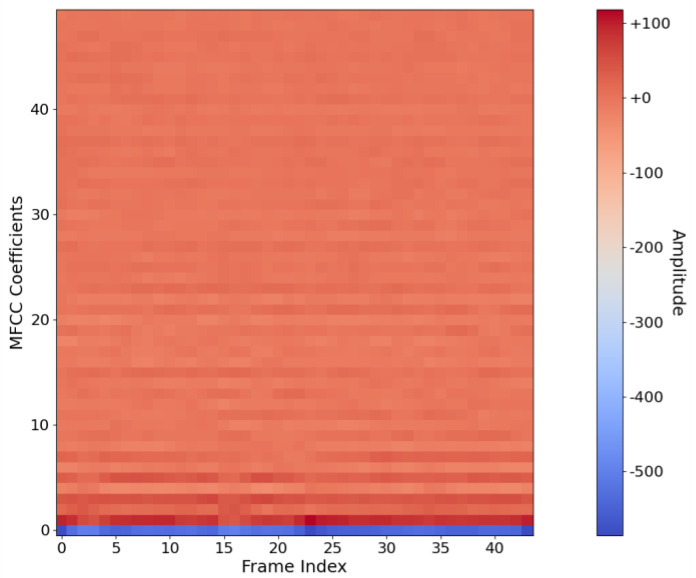
A typical sample of underwater target MFCC feature.

**Figure 3 sensors-24-00273-f003:**
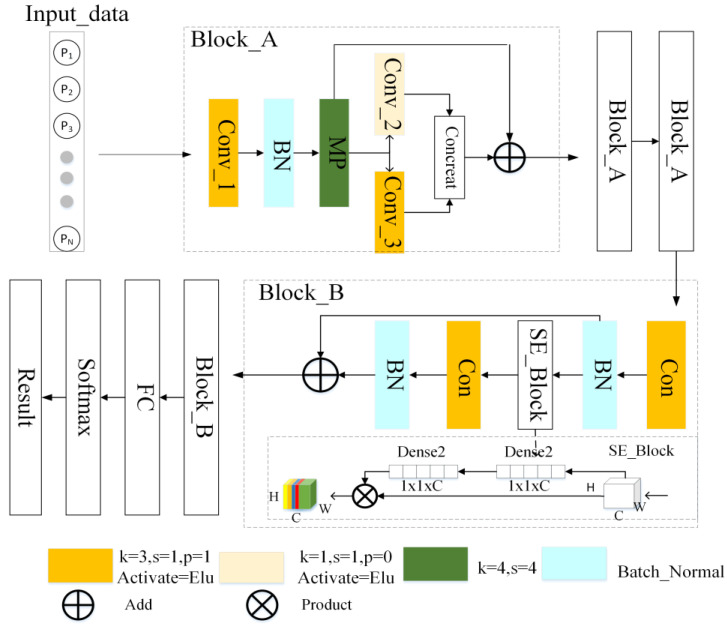
Structure of the proposed RACNN model.

**Figure 4 sensors-24-00273-f004:**
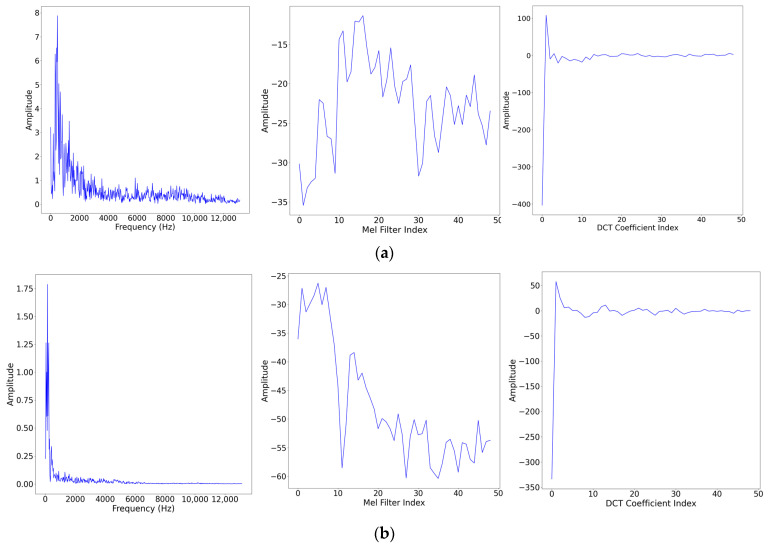
Results of FFT, Mel filter bank, and DCT of different radiated noise. (**a**) Natural noise in Class A. (**b**) Tugboat noise in Class B. (**c**) Motorboat noise in Class C. (**d**) Passengers noise in Class A. (**e**) Ocean liner noise in Class E.

**Figure 5 sensors-24-00273-f005:**
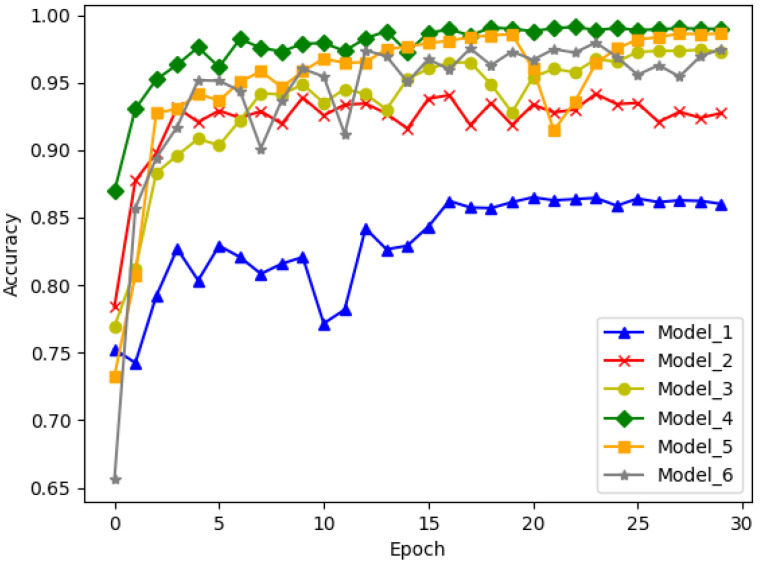
Experimental process for different RACNN models.

**Figure 6 sensors-24-00273-f006:**
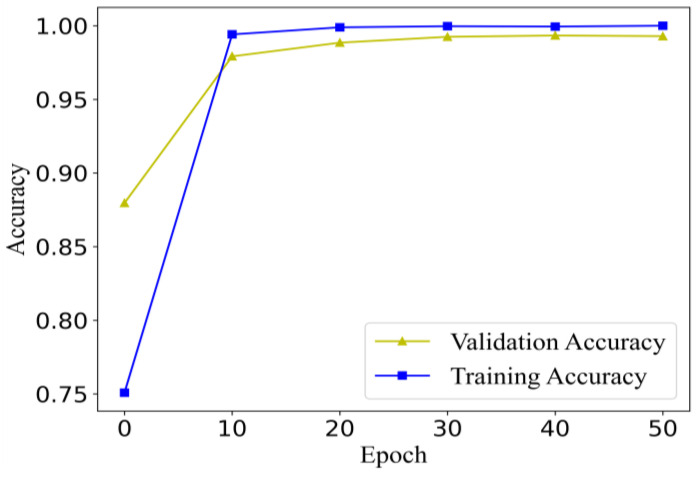
The training process of the proposed RACNN model.

**Figure 7 sensors-24-00273-f007:**
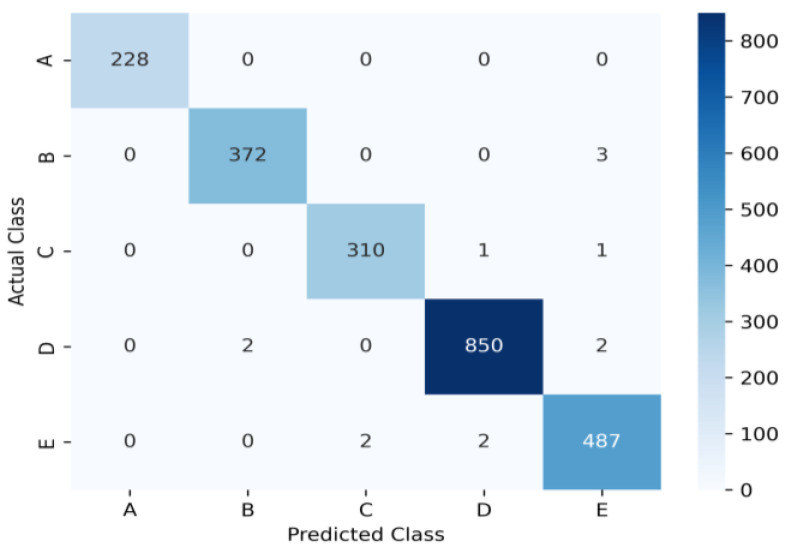
Confusion matrix of RACNN on the test dataset.

**Figure 8 sensors-24-00273-f008:**
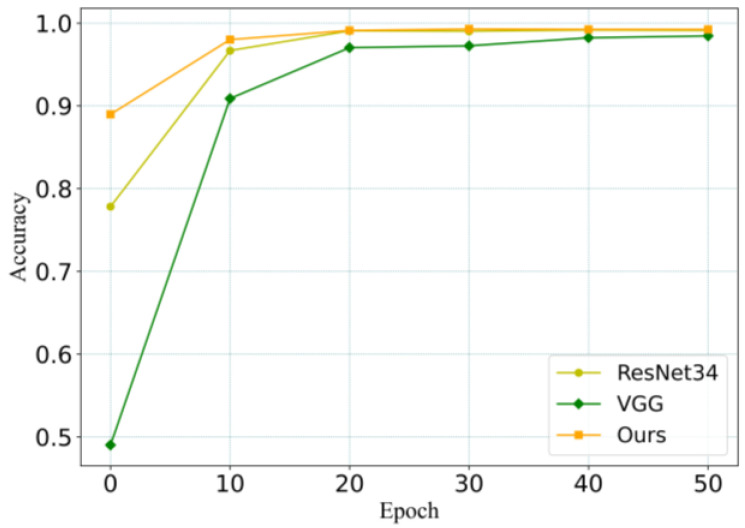
Training process of different deep learning models.

**Table 1 sensors-24-00273-t001:** Dataset for training and testing.

	Class A	Class B	Class C	Class D	Class E	Sum
Train samples Number	912	1500	1248	3416	1964	9040
Test samples Number	288	375	312	854	491	2260

**Table 2 sensors-24-00273-t002:** Experimental results for different RACNN models.

Model	Block_A	Block_B	FC	Acc	Params
Model_1	1	1	None	0.8414	249 K
Model_2	1	1	512	0.9544	18 M
Model_3	3	2	None	0.9646	79 K
Model_4	3	2	256	0.9934	149 K
Model_5	4	3	None	0.9807	99 K
Model_6	4	3	64	0.9783	101 K

**Table 3 sensors-24-00273-t003:** Models with different kernel sizes.

Name	Kernel Size	Activate	Accuracy	Parameter
Model_4	1 × 3	Relu	0.9903	149 K
Model_4	1 × 3	Sigmoid	0.9792	149 K
Model_4	1 × 3	Elu	0.9934	149 K
Model_4	1 × 5	Elu	0.9850	193 K
Model_4	1 × 7	Elu	0.9929	237 K

**Table 4 sensors-24-00273-t004:** Performance of RACNN model on test dets.

Dataset	Class	Precision	Recall	F1-Score	Support
ShipsEar	A	1.000	1.000	1.000	228.0
	B	0.9946	0.9920	0.9933	372.0
	C	0.9936	0.9936	0.9936	312.0
	D	0.9964	0.9953	0.9958	850.0
	E	0.9878	0.9918	0.9898	487.0
	Ave	0.9944	0.9945	0.9945	

**Table 5 sensors-24-00273-t005:** Experimental results of different networks.

Model	Acc	Param
RACNN	0.9934	149 K
Vgg16 [28]	0.9830	33 M
ResNet34 [29]	0.9912	21 M

**Table 6 sensors-24-00273-t006:** Experiment result of different feature models.

Model	Feature	Acc	Param(M)	FLOPs(G)
SVM [30]	STFT	74.6		
1D-CNN [11]	1D-DEMON	94.2	0.4	0.3
AMNet-T [30]	RAW-TIME	97.6	1.69	0.17
CRNN-9 [6]	3D Mel	91.4	0.95	2.57
RACNN	1D-MFCC	99.3	0.149	0.139

## Data Availability

The data presented in this study are openly available in ShipsEar at [27].

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
