# Peer review of "A Novel Underwater Acoustic Target Recognition Method Based on MFCC and RACNN"

_sensors, 2024, doi:10.3390/s24010273_

Round 1
Reviewer 1 Report
Comments and Suggestions for Authors
1、In linelines44-50, the conclusion in of the paper Line needs to be supported by a clear article.
2、In Line85, the MFCC is not a new technology in the paper, where is the innovation.
3、In Line87, what is the point for introducing an attention mechanism to improve CNNs and what is the innovation of the work。
4、In Line Line89, what is the physical significance of reducing paramens.
5、In Line220, the experiment should have a cross-fig for each ship type and the corresponding spectral class.
1、Experiments using RACNN to recognize signals in different frequency bands I feel that it makes little sense, the features are too obvious, or even just use a CNN, what is the feature of using RACNN is not seen. It should be the same frequency band has overlapping signals to show meaningful.
2、The description of the experimental procedure is too weak and the exposition should be enhanced by adding appropriate comparison charts, and noise should be added for testing, how large the dataset is and how large the test data is should be carefully described.
3、Suggest adding real-world data testing。
4、The number of pages in the full text is too low; it is recommended that it be enhanced to 18 pages.
Comments on the Quality of English Language1、In linelines44-50, the conclusion in of the paper Line needs to be supported by a clear article.
2、In Line85, the MFCC is not a new technology in the paper, where is the innovation.
3、In Line87, what is the point for introducing an attention mechanism to improve CNNs and what is the innovation of the work。
4、In Line Line89, what is the physical significance of reducing paramens.
5、In Line220, the experiment should have a cross-fig for each ship type and the corresponding spectral class.
1、Experiments using RACNN to recognize signals in different frequency bands I feel that it makes little sense, the features are too obvious, or even just use a CNN, what is the feature of using RACNN is not seen. It should be the same frequency band has overlapping signals to show meaningful.
2、The description of the experimental procedure is too weak and the exposition should be enhanced by adding appropriate comparison charts, and noise should be added for testing, how large the dataset is and how large the test data is should be carefully described.
3、Suggest adding real-world data testing。
4、The number of pages in the full text is too low; it is recommended that it be enhanced to 18 pages.
Reviewer 2 Report
Comments and Suggestions for Authors
The research introduces a novel method for underwater acoustic target recognition utilizing Mel Frequency Cepstral Coefficients (MFCC) in conjunction with the Residual Attentional Convolution Neural Network (RACNN). This approach effectively captures internal MFCC features from underwater ship-radiated noise. Encouragingly, the experimental outcomes exhibit a commendable overall accuracy of 99.34% on the ShipsEar dataset, surpassing both traditional recognition methodologies and alternative deep learning models.
While the article is well-composed, several suggestions could elevate its quality. Firstly, enhancing the abstract by incorporating a brief discussion of comparable techniques utilized to address the research problem and their associated limitations would be beneficial. Ensuring consistency in keyword usage by incorporating terms like CNN, Deep Learning, MFCC, and appropriately representing the full form of residual and attention mechanisms is recommended. Additionally, structuring the article's organization, particularly by providing an overview of the article's layout at the conclusion of the introduction, would offer readers a clear roadmap of the content.
Moreover, in the introduction section, presenting the related works in a tabulated format for a more accessible comparison with the proposed methodology would enhance comprehension. Concerning Figure 1, clarifying the upper section by including missing text like (FFT, DCT) and adjusting the size and layout of the bottom five graphs to ensure visibility and legibility with a suitable font size is suggested.
Regarding the conclusion section, it should culminate at future work, reevaluating the section to focus on the current methodology's implications and outcomes. It's imperative to maintain consistency in verb tense usage throughout the conclusion, ensuring a cohesive narrative by employing either past or present tense consistently, as the current mixture of tenses may disrupt the section's coherence.
Comments on the Quality of English LanguageMinor editing of English language required
Round 2
Reviewer 1 Report
Comments and Suggestions for Authors
None